**Data Availability Statement:** All relevant data are within the manuscript and its Supporting Information files.

**Funding:** The authors received no specific funding for this work.

# Correlation between neonatal hyperbilirubinemia and vitamin D levels: A meta-analysis

**Jiayu Huang**[ORCID][☯], **Qian Zhao**[☯], **Jiao Li, Jinfeng Meng, Shangbin Li, Weichen Yan, Jie Wang, Changjun Ren**[ORCID]*

The First Affiliated Hospital of Hebei Medical University, Shijiazhuang, Hebei, China

☯ These authors contributed equally to this work.
* 137544907@qq.com

## Abstract

### Objective

Hyperbilirubinemia is a common disease in the neonatal period, and hyperbilirubinemia may cause brain damage. Therefore, prevention and diagnosis and management of hyperbilirubinemia is very important, and vitamin D may affect bilirubin levels. To evaluate the relationship between neonatal hyperbilirubinemia and vitamin D levels.

### Method

The China National Knowledge Infrastructure, VIP, Wanfang, Chinese Biology Medicine Disc, PubMed, Web of Science, Cochrane Library, and Embase databases as well as clinical trial registries in China and the United States were searched for relevant studies from inception to September 2020 without restrictions on language, population, or year. The studies was screened by two reviewers independently, the data were extracted, and the risk of bias of the included studies was evaluated using the NOS. A meta-analysis was conducted on the included studies using Stata11 software.

### Results

Six case-control studies were included, and the methodological quality of the studies was high (grade A). The studies included 690 newborns; more than 409 were diagnosed with hyperbilirubinemia. The means and standard deviations were calculated. Meta-analysis results showed that neonatal vitamin D levels were 7.1 ng/ml lower among infants with hyperbilirubinemia than among healthy newborn levels (z = 6.95, 95% CI 9.10 ~ 5.09, P < 0.05). Subgroup analysis was conducted based on whether the bilirubin levels were concentrated in the 15 to 20 mg/dl range. Vitamin D level of infants with hyperbilirubinemia (the bilirubin levels were concentrated in the 15 to 20 mg/dl range) was 9.52 ng/ml (Z = 15.55, 95% CI-10.72~-8.32, P<0.05) lower than that of healthy infants. The bilirubin levels in four cases were not concentrated in the 15–20 mg/dl range. The results showed that the vitamin D level of hyperbilirubinemia (The bilirubin levels were not concentrated in the 15–20 mg/dl range) neonates were 5.35 ng/ml lower than that of healthy neonates (Z = 6.43, 95% CI-6.98~-3.72, P<0.05).

**Competing interests:** The authors have declared that no competing interests exist.

## Conclusion

Vitamin D levels were observed to be lower in neonates with hyperbilirubinemia as compared to term neonates without hyperbilirubinemia in this study. This can possibly suggest that neonates with lower vitamin D levels are at higher risk for developing hyperbilirubinemia.

## Introduction

Neonatal hyperbilirubinemia a common disease among early newborns; it is also called neonatal jaundice. Approximately 60%-80% of newborns develop jaundice in the first week after birth [1,2]. The main manifestation is yellowish staining of the sclera and skin when the bilirubin exceeds the normal range, especially when the total bilirubin level rises above 5 mg/dl [3]. Bilirubin is decomposed by red blood cells and released directly or produced by hemoglobin derived from red blood cell precursors in the liver and bone marrow and other tissues. Hemoglobin is metabolized by heme oxygenase to produce biliverdin and then converted into bilirubin by the action of biliverdin reductase. The unbound bilirubin is released into the circulation and tightly binds with albumin to form a bilirubin albumin complex. When the complex is transported to the liver, it combines with glucuronidase in liver cells to produce monobilirubin and diglucuronic acid, which are excreted into the bile and intestinal tract. The binding reaction is catalyzed by uridine diphosphate glucuronyl transferase (UGT1A1). In newborns, most of the conjugated bilirubin in the intestine is hydrolyzed back to unbound bilirubin, and this reaction is catalyzed by β-glucuronidase in the intestinal mucosa. Indirect bilirubin is reabsorbed into the bloodstream through the small intestine, forming the enterohepatic cycle. Due to unbound bilirubin is fat soluble which can pass through the blood-brain barrier and the immaturity of the blood-brain barrier during the neonatal period, bilirubin easily accumulates in brain cells, which causes bilirubin encephalopathy and central nervous system insufficiency; this may lead to irreversible damage, especially serum total bilirubin> 20 mg/dl or (and) increasing at a rate of more than 0.5 mg/dl. Therefore, hyperbilirubinemia is one of the most important diseases among newborns; pediatricians should be vigilant and plan early interventions [4–6]. At present, it is generally accepted that the methods for assessing the risk of neonatal hyperbilirubinemia at home and abroad are based on the age of the newborn; total serum bilirubin and/or transcutaneous bilirubin are measured, risk factors are assessed (such as hypoxia, acidosis, head hematoma, sepsis, hypoglycemia), and neonatal jaundice hour bilirubin nomograms are evaluated (Bhutani curve) [7]. There are many complex causes of neonatal hyperbilirubinemia. Common causes include infection, G6PD deficiency, breastfeeding-related jaundice, alloimmunization (ABO hemolysis or rhesus monkey incompatibility, etc.) or other severe hemolysis; there are also numerous unknown factors that lead to hyperbilirubinemia, and these factors need to be further explored [8].

Vitamin D can be dissolved in nonpolar solvents (that is, it is fat soluble). It is a steroid vitamin and can improve bone metabolism. Vitamin D is also involved in calcium and phosphorus metabolism and promotes the normal development of fetal bone marrow cells. Vitamin D is also involved in the proliferation, differentiation and apoptosis of a variety of cells; it has related regulatory effects in the nervous, immune, endocrine and other systems; and it can reduce the incidence of tumors and infectious and allergic diseases [9,10]. Vitamin D is a prohormone. Humans obtain it through food supplementation and endogenous skin synthesis of 7-dehydrocholesterol under sunlight. Vitamin D3 (cholecalciferol) is synthesized through the

skin. It enters the blood and is transported to the liver with vitamin D binding protein (DBP). In food supplements, vitamin D can be in the form of cholecalciferol or ergocalciferol (vitamin D2). Both are absorbed through the lymphatic system as part of chylomicrons, which are metabolized into residual particles, which then deliver vitamin D to the liver. The microsomes in liver cells convert nonactive vitamin D2 and vitamin D3 into 25-hydroxyvitamin D (25-hydroxyvitamin D, 25-OHD) through 25-hydroxase [11]. 25-hydroxyvitamin D is the main and stable form of vitamin D in serum, and its concentration in serum can indicate the body's vitamin D level. 25-OHD produces 1,25-dihydroxyvitamin D under the action of 1-α hydroxylase in the proximal tubule epithelial cells of the kidney, which significantly increases its activity [12].

In addition to participating in the formation of vitamin D, the liver also plays a key role in converting indirect bilirubin into direct bilirubin. Although the metabolic pathways of the two are different, they may affect each other during the biosynthesis stage of the liver. At present, the relationship between vitamin D levels and neonatal hyperbilirubinemia has attracted widespread attention. There have been many epidemiological studies on the relationship between vitamin D levels and neonatal hyperbilirubinemia, and some reports have shown that newborns with hyperbilirubinemia are negatively correlated with their serum vitamin D levels [13–17]. However, there are also studies suggesting that there is no significant correlation between serum vitamin D levels and neonatal hyperbilirubinemia [18]. It is controversial. Therefore, we conducted a meta-analysis of the correlation between neonatal hyperbilirubinemia and serum vitamin D levels to determine whether there is a correlation between the two. The findings will provide evidence regarding the etiology of, risk factors for, and prevention of neonatal hyperbilirubinemia.

## Data and methods

### 1.1 Search strategy

We systematically searched the China National Knowledge Infrastructure (CNKI), VIP, Wanfang, Chinese Biology Medicine Disc, PubMed, Web of Science, Cochrane Library, and Embase databases from inception to September 2020 with no restrictions on language, population or publication year. In addition, we manually searched the reference lists of the included studies to identify additional relevant literature. We also searched the Chinese Clinical Registry and the American Clinical Registry to obtain more unpublished related literature. We used the following combination of subject terms and free terms: "VitaminD"AND"Infants, Newborn"OR"Newborn, Infant" OR"Newborn, Infants" OR"Newborns"OR"Newborn"OR"Neonate"OR"Neonates"AND"Hyperbilirubinemias"OR"Bilirubinemia"OR"Bilirubinemias"OR"Hyperbilirubinemia".

### 1.2 Inclusion criteria and exclusion criteria

Inclusion criteria: ① case-control study; ②study population is newborns (within 1 month of birth); ③ patients in the case group are diagnosed with hyperbilirubinemia; ④newborns in the control group are healthy; ⑤outcome indicator is vitamin D or 25-hydroxyvitamin D; ⑥if the study involving the same population has been published more than once, the study with a larger sample size or with the most recent data was selected; ⑦the study was published in Chinese or English.

Exclusion criteria: ①duplicate publications; ②reviews, systematic reviews, animal experiments; ③inconsistencies in research content, patients or nonpatients with underlying diseases or other inconsistent literature; ④non-case-control studies, unable to obtain the full text, or the outcome indicators do not match or are missing.

### 1.3 Literature screening and data extraction

The literature was screened and the data were extracted independently by two reviewers and cross-checked. If inconsistencies were encountered, they were resolved by discussion. If necessary, the decision was made by a third party. Any missing information was supplemented by contact with the author. The process of literature screening was as follows: exclude the duplicate studies; read the titles and abstracts to exclude irrelevant articles;, and read the full text to identify the included studies. The following data were extracted: first author, publication time, research type, gestational age, gender, birth weight, number of cases and controls, diagnostic criteria of hyperbilirubinemia, source of vitamin D level, mean and standard deviation.

### 1.4 Research quality evaluation

Based on the type of research, we used the Newcastle-Ottawa scale (NOS) to evaluate the quality of the included studies. The following criteria were evaluated: ①is the case determination appropriate; ②representativeness of the case; ③selection of controls; ④determination of controls; ⑤comparability of cases and controls on the basis of the design or analysis; ⑥ascertainment of exposure; ⑦same method of ascertainment for cases and controls; ⑧nonresponse rate. The Newcastle-Ottawa Scale is based on the "star system" and consists of eight items across three areas: choice (four items), comparability (one item) and results (three items). Except for comparable projects, each criterion is awarded one star (comparable projects receive two stars). The highest possible score is 9 points, and the lowest score is 0 points. A score of 7–9 points indicates high methodological quality (level A); a score of 4–6 points indicates moderate methodological quality (level B); and a score of <4 points indicates that the methodological quality is low (level C). The evaluation of the quality of the included studies (the risk of bias evaluation) was carried out by two investigators independently through a blind method, and any differences were resolved through discussion with a third person. Then, any such studies were re-evaluated.

### 1.5 Statistical methods Stata 11.0 software was used for meta-analysis

The measurement data used the mean and standard deviation as the effect statistics and the mean difference and 95% CI were calculated for continuous variables. Five of the included studies reported the mean and standard deviation (SD) of 25(OH)D levels; only one study reported the median and interquartile range (IQR). We tried to get in touch with the author of the latter study to obtain the mean ± standard deviation. Finally, we used a formula to covert the median and IQR into a mean and standard deviation. The heterogeneity among the included research results was analyzed by the Q test (test level is P = 0.1) and the $I^2$ statistic to quantitatively judge the degree of heterogeneity. $I^2$ statistics indicate whether the heterogeneity between studies is low (25–50%), medium (50–75%) or high (>75%). If there was no significant heterogeneity between the results of each study (P>0.1, I2≤50%), the fixed effects model was used for the meta-analysis; if there was significant heterogeneity between the results of the studies (P≤0.1, I2>50%), the random effects model method was used for the meta-analysis after accurately eliminating the adverse effects of obvious clinical heterogeneity. Sensitivity analysis, meta regression and subgroup analysis were used to reduce heterogeneity. The test level of the meta-analysis was p = 0.05. Funnel plots were observed for symmetry to determine whether was publication bias in the outcome.

## Results

### 2.1 The literature search initially yielded 203 relevant articles

Finally, 6 articles were included based on the inclusion criteria and exclusion criteria. The specific literature screening process, reasons for exclusion and results are shown in Fig 1.

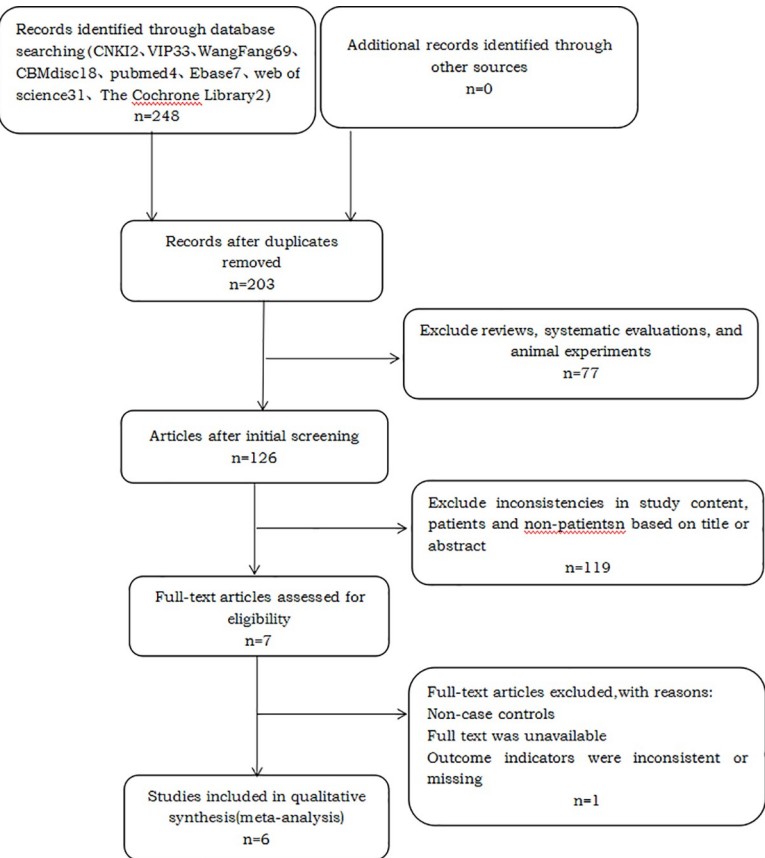

**Fig 1. Flow chart of the search results.**

## 2.2 Characteristics of the included studies

This meta-analysis included 6 articles: 4 case-control studies and 2 prospective case-control studies. Among the 690 newborns included, 409 (59.3%) newborns were diagnosed with hyperbilirubinemia. The bilirubin levels of the children included in the two study case groups were concentrated in the range of 15–20 mg/dl; the bilirubin levels of the children in the other study case groups were not concentrated in this range. The characteristics of the study population are shown in Table 1.

## 2.3 Quality evaluation

The NOS scores of the included studies are shown in Table 2. They were all high-quality articles, including 3 articles with a total score of 8 points and 3 articles with a total score of 7 points.

## 2.4 Meta-analysis results

**2.4.1 Heterogeneity test.** The heterogeneity test ($I^2$ = 72.1%>50%, Q test P<0.1) indicated strong heterogeneity among the studies, and thus, we used the random effects for meta-analysis and investigated the sources of the heterogeneity. Based on the data of this study, it is highly suspected that the source of the heterogeneity was the inconsistent bilirubin level of the case group. Subgroup analysis will be carried out according to the bilirubin level of the case group.

**Table 1. Characteristics of the included studies.**

| Author/Year | research type | characteristics | Diagnostic criteria | Outcome indicators |
|---|---|---|---|---|
| Shao-ting wang 2020 [13] | case-control | Case group 219, sex (male/female) 118/101, gestational age 38.14±1.26, birth weight 2 987.85±382.46 control group 100, sex (male/female) 57/43, gestational age 37.86±2.01, birth weight 3 013.28±429.89 | Practical of Neonatology | 25 (OH)D |
| Elfarargy, M 2019 [14] | case-control | Case group 50, sex (male/female) 30/20, gestational age 39.2±2.1, birth weight 3682±57.3 control group 50, sex (male/female) 31/19, gestational age 39.1±2, birth weight 3698±52.9 | Bilirubin level from 15 to 19 mg/dl on the 3rd day of life in the study group, neonates who required phototherapy according to the American Academy of Pediatrics | 25-OH Vitamin D |
| Bhat, J. A 2019 [15] | Prospective case control | Case group 50, sex (male/female) 30/20, gestational age 38.6±1.5, birth weight 2.75 (0.35) control group 50, sex (male/female) 28/22, gestational age 38.0 ±1.7, birth weight 2.82 (0.43) | Bilirubin levels in the included case group were 15–20 mg/dl | 25- hydroxy vitamin D |
| Shahrokh Mehrpisheh 2018 | case-control | Case group 30, sex (male/female) 17/13, gestational age 38.2 ± 1.3 (37–40), birth weight 3266 ± 365 (2700–3950) control group 30, sex (male/female) 15/15, gestational age 38.4 ± 0.7 (37–40), birth weight 3287 ± 326 (2850–3820) | Increasing of the bilirubin level more than 5 mg/dl was considered as hyperbilirubine-mia. | serum 25-hydroxyvitamin D |
| Aletayeb, S. M. H. 2016 [16] | case-control | Case group 30, sex (male/female) 22/8, gestational age 38–42 week, neonatal weight 3120±495.36 control group 30, sex (male/female) 22/8, gestational age 38–42 week, neonatal weight 3120±353.58 | bilirubin level of >15 mg/dL | 25-hydroxy vitamin D |
| Mutlu, M 2013 [17] | Prospective case control | Case group 30, sex (male/female) 19/11, gestational age 38.2±1.3 (37–40), birth weight 2989±375 (2550–4250) control group 21, sex (male/female) 12/9, gestational age 38.4±0.7 (37–40), birth weight 3256±507 (2150–4300) | a bilirubin level above the preset threshold for phototherapy as recommended by the American Academy of Pediatrics. | 25-OH vitamin D |

For all 6 articles, the random effects model was selected for meta-analysis. The results are shown in Fig 2. The vitamin D level of the case group was 7.1 ng/ml lower than the vitamin D level of the control group, and this difference was statistically significant (Z = 6.95, P<0.05).

**Table 2. Quality assessment of studies.**

| Article | | Selection | Comparability | Exposure | Total points |
|---|---|---|---|---|---|
| author | year | | | | |
| Shao-ting wang | 2020 | 3 | 2 | 3 | 8 |
| Elfarargy, M | 2019 | 3 | 1 | 3 | 7 |
| Bhat, J. A | 2019 | 3 | 1 | 3 | 7 |
| Shahrokh Mehrpisheh | 2018 | 3 | 2 | 3 | 8 |
| Aletayeb, S. M. H. | 2016 | 3 | 1 | 3 | 7 |
| Mutlu, M | 2013 | 4 | 2 | 2 | 8 |

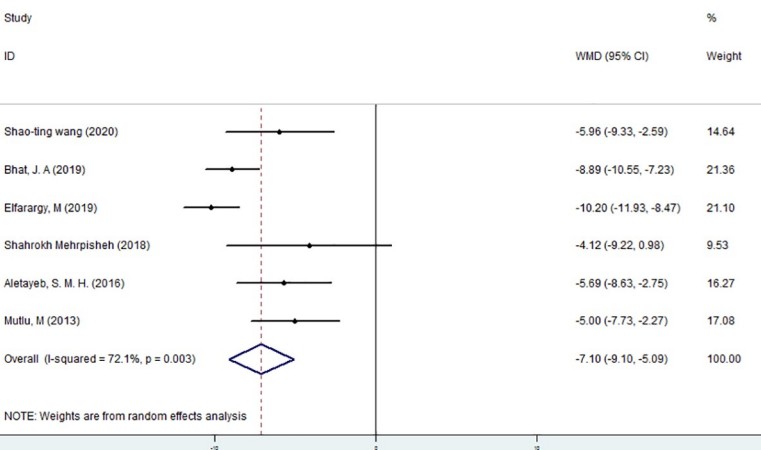

**Fig 2. Meta-analysis of the correlation between vitamin D and neonatal hyperbilirubinemia in general.**

**2.4.2 Sensitivity analysis.** As shown in Fig 3, there are two obvious groupings; that is, the literature shows different sensitivities based on the bilirubin levels of different case groups. Therefore, it is highly suspected that the difference in bilirubin levels among the case groups caused the heterogeneity. Next, we used meta-regression to investigate whether the difference in the bilirubin levels had a significant impact on the effect size.

**2.4.3.** Based on the heterogeneity caused by different bilirubin levels in the case group, we used meta-regression to investigate the source of heterogeneity. The effect size was used as the dependent variable, and the type of bilirubin level in the case group was the independent variable. Table 3 shows that the independent variable "bilirubin level in the case group" had a significant impact on the effect size, P<0.05. Based on this conclusion, a subgroup analysis was conducted.

**2.4.4 Subgroup analysis.** Based on the diagnostic criteria, the six studies were divided into two groups, and meta-analysis was performed. The results are shown in Fig 4. According

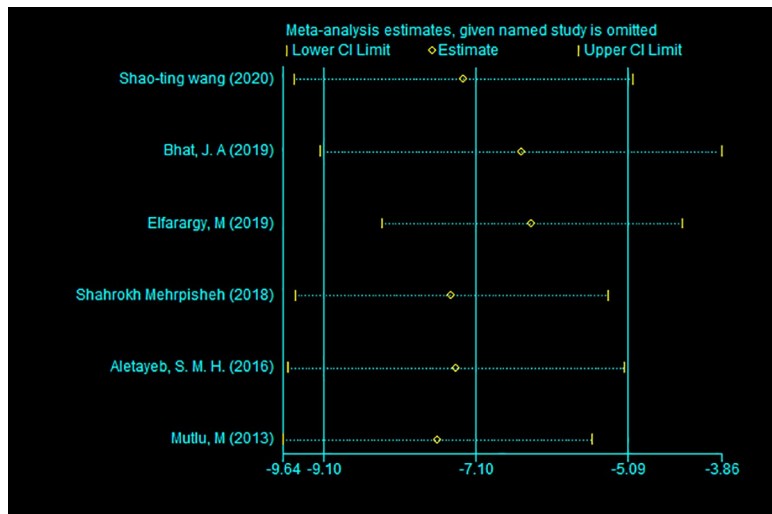

**Fig 3. Sensitivity analysis of included references.**

**Table 3. Meta-regression of different bilirubin levels.**

| _ES | Coef. | Std. Err. | t | P>\|t\| | [95% Conf. Interval] | |
|------|-------|-----------|---|--------|------|------|
| V | 4.171524 | 1.032971 | 4.04 | 0.016 | 1.303537 | 7.039512 |
| _cons | -13.69007 | 1.480178 | -9.25 | 0.001 | -17.7997 | -9.580436 |

to the subgroup analysis, the heterogeneity between the two groups is extremely strong, which means that the diagnostic criteria will largely affect the results of the meta-analysis. The bilirubin level of the two case groups was concentrated in the 15–20 mg/dl range, with no obvious heterogeneity ($I^2$ = 12.5%, P>0.1). The effect size of the subgroup reached -9.52 ng/ml; this effect was statistically significant (z = 15.55 p<0.05), indicating that the vitamin D level of hyperbilirubinemia declined significantly. Second, the bilirubin level in the case group was not concentrated in the 15–20 mg/dl range, and there was no heterogeneity ($I^2$ = 0%, P>0.1). The effect size reached -5.35 ng/ml; this moderate effect was significant (z = 6.43, p <0.05), indicating that the vitamin D level of hyperbilirubinemia strongly decreased. The concentration of bilirubin in the 15–20 mg/dl range for the case group may exaggerate the results. Based on the above analyses, the vitamin D level of newborns with hyperbilirubinemia was significantly lower than that of health newborns.

**2.4.5 Bias test.** Publication bias was assessed based on the subgroups, and a funnel chart was drawn and egger text was done. As shown in Figs 5 and 6, the funnel chart was basically symmetrical, and the egger test had P values greater than 0.05. Therefore, there was no publication bias observed in this meta-analysis.

## Discussion

It is generally believed that vitamin D is essential for the healthy development, growth and maintenance of bones throughout the human body. The main role of vitamin D is to maintain calcium homeostasis. Vitamin D promotes the reabsorption of calcium and phosphorus by the renal tubules and bone matrix through the parathyroid glands, directly affects the reabsorption of calcium and phosphorus in the intestines, and maintains the healthy growth and

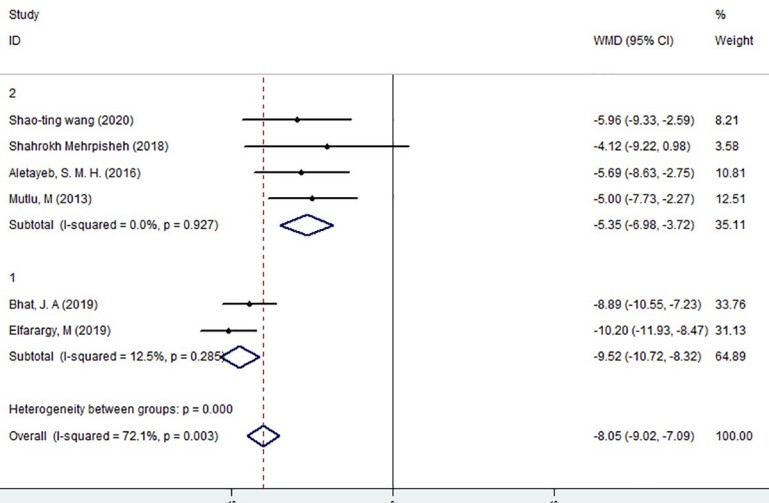

**Fig 4. Meta-analysis of the relationship between vitamin D and neonatal hyperbilirubinemia in different subgroups.**

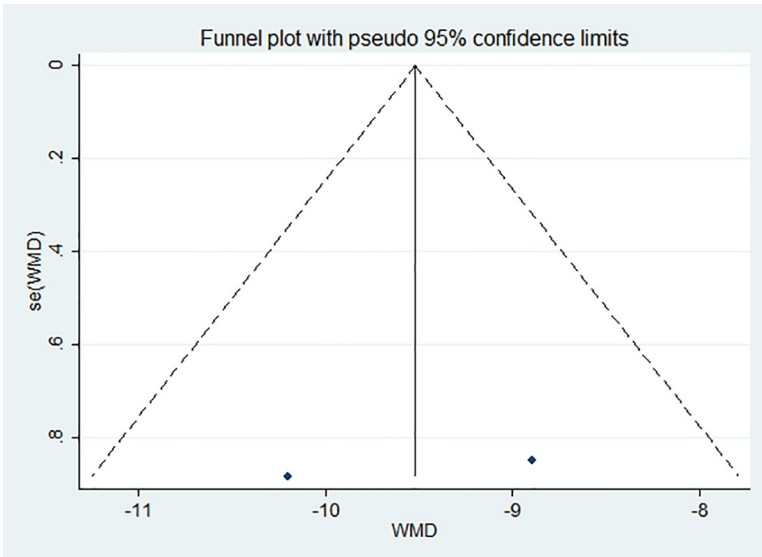

**Fig 5. Publication bias among the papers.** The bilirubin level of the case group was concentrated in the 15–20 mg/dl range is Fig 5.

development of bones under the combined action of calcitonin and parathyroid hormone [19]. At present, the mechanism of the relationship between vitamin D and neonatal hyperbilirubinemia is unclear. The reported biological views are mainly as follows. 1. Indirect blood bilirubin is mostly decomposed by red blood cells. Erythropoietin is the main hormone that promotes red blood cell production. Studies have shown that vitamin D can reduce the level of erythropoietin, so vitamin D deficiency may increase the occurrence of neonatal jaundice [20–22]. 2. Studies have shown that neonatal red blood cells are susceptible to oxidative damage, and vitamin D has certain antioxidant effects, so some people believe that vitamin D helps

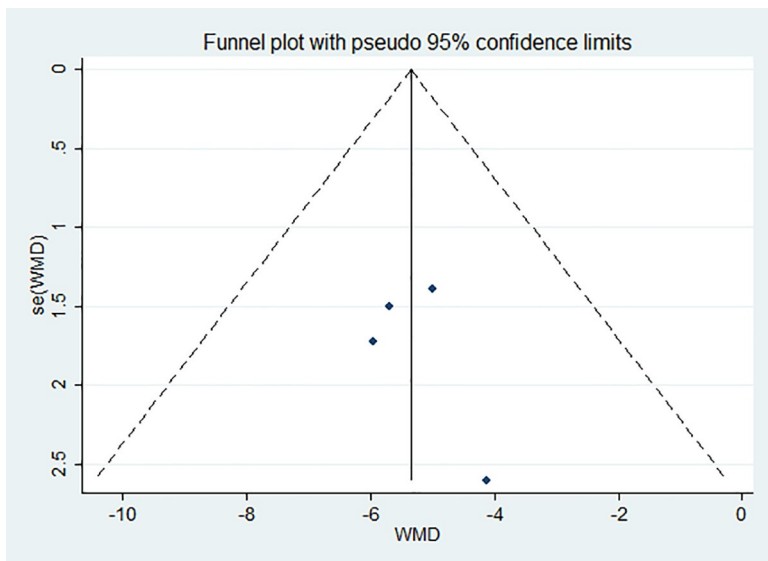

**Fig 6. Publication bias among the papers.** The bilirubin level of the case group was not concentrated in the 15–20 mg/dl range.

prevent the increase in bilirubin levels caused by red blood cell oxidative damage [21,23]. The results of this study show that the vitamin D level of neonates with hyperbilirubinemia is 7.1 ng/ml lower than that of healthy neonates. Subgroup analysis based on the bilirubin level of the study found that the bilirubin level of the case group was concentrated in the 15–20 mg/dl range, which is 9.52 ng/ml lower than that of healthy newborns. This difference in bilirubin values may have been caused by different vitamin D levels, indicating that differences in the degree of neonatal hyperbilirubinemia are caused by different vitamin D levels. At present, only one study has reported finding no significant difference in serum 25-(OH)D3 levels among neonates with different severities of hyperbilirubinemia [24], but more clinical studies are needed to support this hypothesis.

The strengths of this study are as follows. First, the included studies were all case-control experiments, and their quality was high, which reduced bias and improved the reliability of the outcomes. Second, all of the included studies corrected for potential confounders, which enhanced the credibility of the results. Third, we found the sources of the overall heterogeneity and conducted subgroup analysis. The subgroup analysis reduced the heterogeneity between the groups, and the results were the same as those of the overall meta-analysis. This study also has certain limitations. First, the number of included articles is small, which may have had some impact on the quality of the research results. In the future, more studies will need to be included. Therefore, we conducted rigorous analyses before interpreting the results. Second, four of the six included studies were from Asian populations. Different populations may have different results and are more likely to produce contradictory outcomes. Future research should ensure that different populations are studied.

We scientifically evaluated the relationship between vitamin D levels and neonatal hyperbilirubinemia with a meta-analysis. The quality of the included studies was evaluated with the NOS. The results showed that the quality of each study was high. The results of the meta-analysis showed that the vitamin D level of the case group was significantly lower than the vitamin D level of the control group(P<0.05). The source of overall heterogeneity was found through meta-regression and subgroup analysis, the results of which were found to be consistent with the overall meta-analysis. It was found that among the subgroups, the differences of the group in which patients bilirubin level was concentrated in the 15–20 mg/dl was more obvious.

Above all, vitamin D levels are correlated with the occurrence of neonatal hyperbilirubinemia. Insufficiency or lack of vitamin D may be one of the risk factors for neonatal hyperbilirubinemia. Some studies have shown that adding vitamin D to pregnant women's diets is associated with a decrease in neonatal hyperbilirubinemia. This indicates that vitamin D is important in reducing bilirubin levels in jaundice neonates. In other words, the vitamin D levels of newborns with jaundice are low. These findings also suggest that mothers should take vitamin D to reduce the level of bilirubin in newborns [25]. Therefore, timely and effective vitamin D supplementation for pregnant mothers and newborns can reduce the risk of neonatal hyperbilirubinemia. However, there are few studies on the mechanism of the relationship between vitamin D levels and the risk of neonatal jaundice. This conclusion still needs to be further verified by more large-sample, high-quality, multiethnic studies.

## Supporting information

**S1 Checklist. PRISMA 2009 checklist.**
(DOC)

**S1 File.**
(ZIP)

## Author Contributions

**Conceptualization:** Jiayu Huang, Qian Zhao.

**Data curation:** Jiao Li, Jinfeng Meng.

**Formal analysis:** Jiayu Huang, Qian Zhao.

**Investigation:** Jiayu Huang, Qian Zhao, Shangbin Li.

**Methodology:** Jiayu Huang, Qian Zhao, Weichen Yan.

**Supervision:** Weichen Yan, Jie Wang, Changjun Ren.

**Validation:** Jiao Li, Jinfeng Meng, Shangbin Li.

**Visualization:** Jie Wang, Changjun Ren.

**Writing – original draft:** Jiayu Huang, Qian Zhao.

**Writing – review & editing:** Jiayu Huang, Qian Zhao, Changjun Ren.

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
