## [Decision Letter · Decision Letter 0]

30 Mar 2021

PONE-D-20-33494

Correlation between neonatal hyperbilirubinemia and vitamin D levels: A meta-analysis

PLOS ONE

Dear Dr. Ren,

Thank you for submitting your manuscript to PLOS ONE. After careful consideration, we feel that it has merit but does not fully meet PLOS ONE’s publication criteria as it currently stands. Therefore, we invite you to submit a major revision to  version of the manuscript that addresses the points raised during the review process by both reviewers.

We look forward to receiving your revised manuscript.

Kind regards,

Abhishek Makkar, M.D.

Academic Editor

PLOS ONE

Journal Requirements:

2. At this time, we ask that you please provide the full search strategy and search terms for at least one database used as Supplementary Information.

3. Please confirm that you have included all items recommended in the PRISMA checklist including details of reasons for study exclusions in the PRISMA flowchart and number of studies excluded for each reason.

Reviewers' comments:

Reviewer's Responses to Questions

**Comments to the Author**

1. Is the manuscript technically sound, and do the data support the conclusions?

Reviewer #1: Yes

Reviewer #2: No

2. Has the statistical analysis been performed appropriately and rigorously? 

Reviewer #1: I Don't Know

Reviewer #2: Yes

3. Have the authors made all data underlying the findings in their manuscript fully available?

Reviewer #1: Yes

Reviewer #2: Yes

4. Is the manuscript presented in an intelligible fashion and written in standard English?

Reviewer #1: Yes

Reviewer #2: No

5. Review Comments to the Author

Reviewer #1: Comments and suggestions for the authors:

This is an interesting meta-analysis on neonatal hyperbilirubinemia and Vitamin D levels.

1) Suggestions on Abstract: The abstract submitted does not have a purpose or background. Suggested to include few lines on neonatal hyperbilirubinemia and why it is important to identify if there is any correlation between the two.

2) For the results stated in the abstract and in the main paper, would recommend to include the SI units for the vitamin d levels. Were the levels of Vitamin D reported in all the papers with the same metric measurements?

3) Suggestions to modify the conclusion for the abstract: Vitamin D levels were observed to be lower in neonates with hyperbilirubinemia as compared to term neonates without hyperbilirubinemia in this study. This can possibly suggest that neonates with lower vitamin D levels are at higher risk for developing hyperbilirubinemia

4) Corrections:

Line 42: A meta-analysis was conducted on the included studies using Stata11.0 43 software.

Line 48: Vitamin D level of infants with hyperbilirubinemia (15 to 20 mg/dl group?) was 9.52 (Z =15.55, 95% CI-10.72~-8.32, P<0.05) lower than that of healthy infants.

Line 50: It is unclear what these results are? vitamin D level of hyperbilirubinemia neonates were 5.35 lower than that of healthy neonates (Z =6.43, 95% CI-6.98~-3.72, 51 P<0.05).

5) Introduction: Introduction can be more precise on neonatal hyperbilirubinemia and vitamin D levels. More on why the correlation needs to be done. The introduction of the manuscript is very elaborate

6) When the number of studies included in the analysis is less than 10, the Hartung-Knapp-Sidik-Jonkman method can better reduce the risk of type 1 error than the DerSimonian and Laird method. What method did the authors use for the random effect?

7) Funnel plot is recommended only when more than 10 articles are included in the meta-analysis. Can authors explain why they chose to subgroup and do a funnel plot for only two studies and four studies?

Reviewer #2: The main concern about this article is it does not describe the units properly. For eg, it states: The vitamin D level of infants with hyperbilirubinemia was 9.52 (Z =15.55, 95% CI-10.72~-8.32, P<0.05) lower than that of healthy

infants. What is 9.52? is it 9.52 times, is the vitamin D level 9.52, if so in what units? The way it is written it is very hard to understand. Also of the articles selected for the metanalysis, no description is given of how they measured vitamin D levels. Generally, there is considerable variation in the assays used with mass spec being the most accurate.

6. PLOS authors have the option to publish the peer review history of their article (what does this mean?). If published, this will include your full peer review and any attached files.

Reviewer #1: No

Reviewer #2: **Yes: **SOWMYA KRISHNAN

---

## [Author Response · Author response to Decision Letter 0]

11 Apr 2021

Dear Reviewers:

 Thank you for your letter and for the Editors and reviewers’ comments concerning our manuscript entitled ”Correlation between neonatal hyperbilirubinemia and vitamin D levels: A meta-analysis”(ID: PONE-D-20-33494). Those comments are all valuable and very helpful for revising and improving our paper, as well as the important guiding significance to our researches. We have studied comments carefully and have made correction which we hope meet with approval. Revised portion are marked in red in the paper. 

Once again, thank you very much for your comments and suggestions. 

 Yours sincerely,

Changjun Ren

---

## [Editor Report · Decision Letter 1]

23 Apr 2021

PONE-D-20-33494R1

Correlation between neonatal hyperbilirubinemia and vitamin D levels: A meta-analysis

PLOS ONE

Dear Dr. Ren,

Thank you for submitting your manuscript to PLOS ONE. After careful consideration, we feel that it has merit but does not fully meet PLOS ONE’s publication criteria as it currently stands. Therefore, we invite you to submit a revised version of the manuscript that addresses the points raised during the review process.

Thanks for addressing reviewer comments. I would advise you to change term judgement in second line of abstract and replace with diagnosis and management. Judgement isn't a medical term and isn't clear what  you are trying to state. Looking forward to your final version.

We look forward to receiving your revised manuscript.

Kind regards,

Abhishek Makkar, M.D.

Academic Editor

PLOS ONE
---

## [Editor Report · Decision Letter 2]

29 Apr 2021

Correlation between neonatal hyperbilirubinemia and vitamin D levels: A meta-analysis

PONE-D-20-33494R2

Dear Dr. Ren,

We’re pleased to inform you that your manuscript has been judged scientifically suitable for publication and will be formally accepted for publication once it meets all outstanding technical requirements.

Kind regards,

Abhishek Makkar, M.D.

Academic Editor

PLOS ONE
---

## [Editor Report · Acceptance letter]

4 May 2021

PONE-D-20-33494R2 

Correlation between neonatal hyperbilirubinemia and vitamin D levels: A meta-analysis 

Dear Dr. Ren:

I'm pleased to inform you that your manuscript has been deemed suitable for publication in PLOS ONE. Congratulations! Your manuscript is now with our production department. 

Kind regards, 

on behalf of

Dr. Abhishek Makkar 

Academic Editor

PLOS ONE